# OpenReview forum: "Solving Math Word Problems with Process-based and Outcome-based Feedback"
_ICLR.cc/2023/Conference — Submitted to ICLR 2023_

### Official Review · Reviewer_GLbN · 2022-10-24

**Confidence:** 4
**Clarity, Quality, Novelty And Reproducibility:** The paper is clear and well presented.
**Correctness:** 3
**Technical Novelty And Significance:** 2
**Empirical Novelty And Significance:** 2
**Recommendation:** 3

**Strength And Weaknesses:**

### strengths
I believe the paper is on an important direction. A systematic evaluation of different approaches for MWP solving is very much appreciated and needed.


### weaknesses
I wish the results and analyses can go a bit deeper than what have been presented. The main 4 results appear in the paper multiple times and, along with the detailed experiment setup (which I appreciate), can be more succinct (or move some to appendix to highlight the major findings). The authors investigations seem to center around two approaches to fine-tune the model (e.g., output vs. process-based rewards). I think a few other settings can also be considered, such as whether there is a prompt for fine-tuning/inference (similar to few-shot) and the number of examples used in the prompt. Or compare different decoding schemes.

As a scientific investigation paper, I expect deeper analyses, in addition t o simply presenting the observations, to enable finding more insights (at present, most of the results in the paper are somewhat expected). For example (a few suggestions):
- ORM and PRM give roughly the same final-answer correctness. Could the authors design an experiment to investigate why this is the case, e.g., empirical evidence to support the hypothesis it is "simpler for ORM to recognize when steps are correct than it is to check the answer by internally computing the final answer itself"?
- The authors claim ORM seems to learn to emulate the PRM. How does it do that, even without explicitly being supervised on the correctness of each intermediate step?

**Summary Of The Paper:**

The paper investigates different approaches to solve math word problems (MWPs), specifically on the GSM8K dataset. The authors compare training with 1) final answer correctness only (ORM), 2) each intermediate step's correctness (PRM), and 3) with different reward models for ORM and PRM. The authors find that 1) ORM and PRM achieve similar final answer correctness; 2) PRM and PRM fine-tuned models learn to emulate step-by-step generated solutions; 3) PRM (or similar approaches) is required to improve the correctness of each intermediate solution step; and 4) allowing the model to abstain significantly improves performance (on the not-abstained solutions).

**Summary Of The Review:**

The depth (both technical and experimental aspects) of the paper feels more like a workshop paper (or can be written more succinctly to summarize the main findings). If the authors intend to present their work as a scientific investigation, I encourage the authors to more systematically consider the scientific questions and experimental evaluations. I look forward to deeper insights that the authors can bring on the different approaches that current/existing methodologies are tackling MWP solving and how these approaches compare to each other.

---

> ### Author Response · Authors · 2022-11-09
> **Reply to reviewer GLbN**
>
>
> > The authors investigations seem to center around two approaches to fine-tune the model (e.g., output vs. process-based rewards). I think a few other settings can also be considered, such as whether there is a prompt for fine-tuning/inference (similar to few-shot) and the number of examples used in the prompt. Or compare different decoding schemes.
>
> We tried to focus our experiments on settings which we thought might change the relationship between process-based and outcome-based supervision which is how we decided on the set of variations that we included in the paper.  While we feel these are the most likely to make a difference on the results, we can include specific additional variations if you believe they are important in this comparison. Regarding the specific suggestions:
>
>
>
> *   Prompt required for fine-tuning/inference:  In general when there’s a large amount of fine-tuning data (as there is for GSM8K), our experience has been than prompting on top of this doesn’t make much of a difference so we didn’t include any results along this vein.
> *   Number of examples used in the prompt: All the few-shot methods substantially underperform finetuned approaches, regardless of the number of prompt examples.
> *   Different decoding schemes:  We agree that this is interesting, and we have a broader comparison of decoding schemes in Table 3 in the Appendix.
>
> In general, we felt we were already exploring many axes of variation, so wanted to simplify as much as possible. That said, if there are particular configurations or dimensions which seem particularly interesting to you, we would be happy to include them.
>
> >  I expect deeper analyses, in addition to simply presenting the observations, to enable finding more insights (at present, most of the results in the paper are somewhat expected).
>
> As discussed above, we believe our main 4 findings are both surprising and insightful. We aren’t claiming they’re totally surprising (no one would set out to do any experiments if they didn't expect them to work out based on some prior experience), but we think most experts would not have correctly predicted the full results we’ve presented. We’d appreciate discussing further, either here or above.
>
> > ORM and PRM give roughly the same final-answer correctness. Could the authors design an experiment to investigate why this is the case, e.g., empirical evidence to support the hypothesis it is "simpler for ORM to recognize when steps are correct than it is to check the answer by internally computing the final answer itself"? The authors claim ORM seems to learn to emulate the [process-based labels]. How does it do that, even without explicitly being supervised on the correctness of each intermediate step?
>
> We agree this is interesting, and this was the motivation for the analysis in Figure 4. The most direct empirical evidence is what we provide: the reason to suspect this behavior is “simpler” is that empirically, this is the behavior learned by the ORM. Of course, it is difficult to ever explain _why_ neural networks learn one behavior or another, or what gradient descent prefers, or numerous other choices that affect this. While this is an active area of research, we aren’t aware of any methodology which can answer this question without being an entire research paper on its own. If you are aware of any methodology we could apply to answer this questions, please let us know.

---

### Official Review · Reviewer_pkRc · 2022-10-25

**Confidence:** 3
**Correctness:** 3
**Technical Novelty And Significance:** 2
**Empirical Novelty And Significance:** 2
**Recommendation:** 3

**Clarity, Quality, Novelty And Reproducibility:**

The paper is well written. I think the lack of clarify comes from the selection of results. The work is novel as organized experiments using mostly existing methods.

Questions:
- It is possible to solve the problem with actions in a different order. If so, does that affects the claim of the paper?


**Strength And Weaknesses:**

Strengths
- Scholarship on methods for training for correctness.
- Some combinations of methods produced significant improvements.
- Good attempt to synthesize the empirical observations

Weaknesses
- The organization of sections 2.3 to 2.5 is a bit confusing. Perhaps there are too many combinations in each block of Table 1. Even when I attempt to re-read, I remain confused about how the vertical caption "RM rerank" is related to the RM using majority, ORLM or PRM.
	- The legend of Fig 3 is also revealing. There are three axis combines in partial combinations, but only some combinations appear.
	- I'd consider working backwards from the findings, and reducing the number of experiments to show. That might imply moving some results to the appendix and keeping only short comments in the main body on how other methods give similar results.
- I find it hard to know if the multiple decisions/hyper-parameters per method could be affecting the validity of the results.
- The observation that the models get away with correctness while failing at intermediary steps might be explained by an easier chance to introduce artifacts/shortcuts that improve the accuracy.


**Summary Of The Paper:**

The papers study the finetuning of LLMs for solving maths problems. In particular, comparing training partial vs overall correctness. While the evaluation shows that both produce similar global accuracy, partial correctness requires feedback of that kind. The paper considers multiple existing alternatives for both kinds of signals. A group of them manage to get significant improvement, but there is not a clear combination that helps, perhaps because some of the methods are similar.

**Summary Of The Review:**

The paper considers the kind of supervision for MATH reasoning problems using LLM. The line of work is important, but I'm afraid it's hard to get concrete lessons from the paper.

---

> ### Author Response · Authors · 2022-11-09
> **Reply to reviewer pkRc**
>
>
> > The organization of sections 2.3 to 2.5 is a bit confusing. Perhaps there are too many combinations in each block of Table 1. Even when I attempt to re-read, I remain confused about how the vertical caption "RM rerank" is related to the RM using majority, ORLM or PRM.
>
> Thanks for the great feedback:
>
>
>
> *   RM (ORM or PRM) reranking is mutually exclusive with majority voting (Section 2.5 / Figure 1). If the method doesn’t use an RM, then it is using majority voting (second vertical block). Otherwise, it’s either using ORM or PRM for reranking, as specified in each row.
> *   We have added majority voting explicitly to the approach names used in the table. Does that clarify the table for you?
>
> > The legend of Fig 3 is also revealing. There are three axis combines in partial combinations, but only some combinations appear.  I'd consider working backwards from the findings, and reducing the number of experiments to show. That might imply moving some results to the appendix and keeping only short comments in the main body on how other methods give similar results.
>
> We’ve addressed this point in our overall response.
>
> > I find it hard to know if the multiple decisions/hyper-parameters per method could be affecting the validity of the results.
>
> This is a valid concern, but we are confident about our results. In general, we’ve selected hyperparameters based on the SFT experiment (and some slight tuning of the ORM training, though that mostly reuses SFT hyperparameters). Then, given our RL training also reduces to SFT, there isn’t much room for hyperparameter tuning on top. It’s true there are a couple more free variables (e.g. how to initialize the RL policy), where we didn’t sweep, but this will always be true to some extent. Importantly, the effect sizes for all our key results are very large, and so should be robust.
>
> Additionally, with our SFT hyperparameter tuning, SFT results are fairly robust across hyperparameters (variation in SFT + majority voting across runs is maybe 1%).

---

> > ### Comment · Reviewer_pkRc · 2022-11-22
> > **Thank you**
> >
> > Thank you for your comments. Perhaps our concern with novelty is just that we are not understanding the paper.
> > I hope the feedback would make this work into a stronger paper. It's an important problem

---

### Official Review · Reviewer_ytw5 · 2022-10-26

**Confidence:** 4
**Correctness:** 3
**Technical Novelty And Significance:** 2
**Empirical Novelty And Significance:** Not applicable
**Recommendation:** 5

**Clarity, Quality, Novelty And Reproducibility:**

The paper is well-organized and easy to follow. However, the paper appears to have made only a minor contribution to modeling novelty.

**Strength And Weaknesses:**

**Strengths**

The paper comprehensively compares process- and outcome-based approaches on the math word problem dataset, GSM8K. It reports many impressive results of outcome-based and process-based approaches on final-answer error rates and trace errors. These results are meaningful for the follow-up work to develop models with both low error rates and trace errors. Figure 3 is very informative and shows that the trace errors are associated with final answer errors, especially when final answer errors are low.


**Weaknesses/Feedback**

**1. The methods have limited modeling novelty.**

It is good to see that the paper conducts the *first* comprehensive comparison between process- and outcome-based approaches trained on the math word problem-solving task. The paper conducts extensive experiments on the two types of approaches, and there are a lot of interesting findings. However, it seems that the paper does not propose a new method.

**2. All the experiments are conducted on only one dataset.**

I understand that there are some limitations for other datasets, and conducting experiments on new datasets leads to extra costs. However, whenever the annotation pipeline is set up, the additional costs of conducting experiments on new datasets are acceptable. It is worthwhile to conduct the experiments on smaller datasets. A recently proposed MWP dataset, TabMWP (https://arxiv.org/abs/2209.14610), might be a good benchmark for the experiments in the revised paper, which is annotated with multi-step reasoning steps.


**Summary Of The Paper:**


Most of the recent work on language models for solving math word problems relies on prompting engineering. Instead, this paper studies how to finetune the language models for solving math word problems with outcome-based and process-based approaches. The paper develops two metrics: race error rate and final-answer error rate. The paper implements multiple outcome-based and process-based approaches and conducts experiments on a recently proposed dataset, GSM8K. The best model could improve the previous best results from 16.8% to 12.7% final-answer error and from 14.0% to 3.4% trace error.


**Summary Of The Review:**

The paper is interesting to me, but there are two main concerns: the methods have limited modeling novelty, and all the experiments are conducted on only one dataset. It would be helpful to see revisions regarding these two concerns.

---

> ### Author Response · Authors · 2022-11-09
> **Reply to reviewer ytw5**
>
>
> Thank you for the thoughtful, detailed review and suggestions, and particularly for the pointer to TabMWP.
>
> > 1.  The methods have limited modeling novelty
>
> As discussed above, we agree, but do not view this as a shortcoming.
>
> > 2. All the experiments are conducted on only one dataset.
>
> As per above, we’ve conducted some preliminary experiments on TabMWP, and could do more if it’s of interest. Two further details:
>
>
>
> *   In our experience, a small but significant amount of (task-specific) training made a big difference in annotator performance. Since we don’t want to compare to poorly implemented baselines, expanding to non-MWP tasks (e.g. CommonsenseQA, or HotpotQA) would likely require additional setup work. _That said_, TabMWP looks like a nice dataset  (if it’d been available during the work, we may have included it), and it’s also in the MWP domain, so would be feasible for us to include.
> *   One wrinkle with TabMWP is that many of the current mistakes are calculation mistakes. We believe we could train the model to use a calculator on TabMWP using either a process-based (directly augmenting the SFT dataset) or outcome-based (e.g. few-shot ORM-RL) approach. However, after doing this, we would expect performance to be quite high for both process- and outcome-based approaches, so it may be hard to draw meaningful comparisons. Would this still be of interest?

---

### Author Response · Authors · 2022-11-09
**To all reviewers (1/2)**

To all the reviewers,

Thank you for the thoughtful, detailed reviews and suggestions. We address here the major concerns of all reviewers while we directly respond to more detailed concerns of each reviewer in our individual reviewer responses.

**Novelty**

Reviewer ytw5 is concerned that “The methods have limited modeling novelty” while reviewer GLbN finds that most of the results are “somewhat expected”.

We believe these should **not** be considered major concerns for the following reasons:



1. The focus of our work is on novel findings rather than novel methods:
    1. The ICLR policy asks “[does it present novel findings](https://iclr.cc/Conferences/2023/ReviewerGuide#Reviewing%20instructions)”, rather than limiting exclusively to methods. It also asks "Does it contribute new knowledge" and says "Submissions bring value to the ICLR community when they convincingly demonstrate new, relevant, impactful knowledge (incl., **empirical**, theoretical, **for practitioners**, etc)".
    2. Our work provides important empirical contributions. While there’s been a plethora of work in this space focused on prompting, no other work has done a comparison of fine-tuning techniques despite the fact that finetuning leads to significantly better empirical results. A systematic comparison of finetuning methods using the same base model and task is important for the community to aid understanding of their relative strengths and weaknesses.
    3. Additionally, the novelty of our empirical results are highlighted by reviewer pkRc “The work is novel as organized experiments using mostly existing methods.” and reviewer GLbN “A systematic evaluation of different approaches for MWP solving is very much appreciated and needed.”
2. We believe that our main findings are both surprising and impactful. Specifically:
    1. **Supervising final-answer correctness alone suffices for low final-answer error rate:** Prior work on GSM8K has not done a direct comparison between finetuning with full supervision (i.e. including the reasoning traces) and fine-tuning with only final answer supervision. We expected that supervision on the reasoning traces would lead to significantly improved performance and were quite surprised to find that the final-answer errors were almost equivalent in these two setups.
    2. **Outcome-supervised reward models approximate process-based labels:**  This was the most surprising finding to us, as the default expectation would be for the outcome-supervised model to approximate the outcome-based labels, rather than the process-based labels.
    3. **Pure outcome-based feedback leads to high trace error.** One of the most frequent questions about various outcome based results (specifically Few-shot+Final Answer RL from Zelikman, 2022) is whether or not the intermediate traces are meaningful.  Our paper is the first to look at the effect of different modeling choices on trace accuracy. While it may be somewhat expected that process-based feedback leads to lower trace-error, we found it interesting that the gap in trace errors was so significant.  We expected some difference, but 11.4% vs 19.8% (\~2x), and 4.4% vs 12.4% (\~3x) were larger margins than we expected.
    4. **Low trace error can be attained using an outcome-supervised reward model (emulating process-based feedback).** Given that the outcome-supervised RM is trained to approximate the final-answer correctness, it is initially surprising that it so dramatically reduces trace error (12.4% down to 5.5%) compared to optimizing final-answer correctness, leading to a trace accuracy for the outcome supervised model which is only a couple of percent higher than the process supervised model (5.5% vs 3.4%). This makes sense given the RM analysis referenced above, but without that, seems unexpected (and was largely the reason we investigated the RM behaviors more closely).
    5. **Selective prediction greatly reduces final-answer error, particularly for models with low trace error.** While this may not be entirely surprising in itself, this result has important implications for deployment of such models in domains like education where error rates above a few percent are typically not acceptable.
3. Finally, while different people may disagree about what is and isn’t surprising, we note that despite GSM8K being a highly active benchmark (76 cites over past year, and the main benchmark used by verbalized reasoning/chain-of-thought papers), nobody has tried our best-performing approaches on GSM8K before. We significantly improve state-of-the-art despite starting with a weaker base model.

---

> ### Author Response · Authors · 2022-11-09
> **To all reviewers (2/2)**
>
>
> **Additional datasets**
>
> We thank reviewer ytw5 for the pointer to the TabMWP dataset. We’ve run an initial experiment on the dataset, and achieved an accuracy of 89% using (process-based, with reasoning traces) SFT, compared to (as far as we know) a previous SOTA of 70.9% using InstructGPT with GPT-3. We believe that we could improve this to around 95% or more by combining this with Few-shot ORM-RL and a calculator, since the TabMWP demonstrations do not demonstrate calculator use, but few-shotting combined with RL can quickly add this ability to the model. However, while it may be possible for us to do this, it would be a significant amount of work, so we’d like to check with all reviewers whether this would change your evaluation, before we do this extra work. Reviewers pkRc and GLbN (in addition to reviewer ytw5): is this an addition that would lead you to increase your score?
>
> **Reporting more vs. fewer approaches**
>
> Reviewer GLbN suggests that “other settings can also be considered”, while Reviewer pkRc suggests “reducing the number of experiments to show”:
>
>
>
> *   We’ve structured Table 1 around supporting the 4 key results we highlight in the Results section.
> *   We could include fewer results, but:
>     *   it’s helpful to see trends across settings (e.g. process-based feedback significantly reduces trace error, both with and without a RM)
>     *   we wanted interested readers to be able to see the full results.
> *   We could include more results (e.g. the full cross-product, or additional axes of variation), but that would mean even more for the reader to process.
>     *   The reason we don’t include a full cross-product is that within each setting (no RM, RM for reranking only, RM for RL and reranking), we are varying between process vs. outcome-based feedback, and leaving everything else fixed.
> *   Overall, we felt like the set of results we included was a good balance, but we’ve made some tweaks, and are open to any specific suggestions of what to include or remove.  We could, for example, remove the PRM-RL results since they are largely similar to the ORM-RL results if the reviewers felt like that would be a helpful change.
> *   We agree that the results in Table 1 are hard to interpret in isolation and so we’ve updated the caption to point directly to the findings outlined in the results section.

---

> > ### Comment · Reviewer_pkRc · 2022-11-22
> > **Consensus**
> >
> > Hi, I think we have a consensus around the decision, even if the decision is not for the right reasons.
> > I suspect we are not receiving the contributions, and we are diverging on the reasons.
> >
> > I hope the reviews help the authors to improve the paper.

---

### Decision · Program_Chairs · 2023-01-20

**Decision:**

Reject

**Justification For Why Not Higher Score:**

None of the reviewers would like to recommend acceptance in its current form. While I appreciate the authors' efforts in clarifying questions, some critical questions remain unclear, and I don't feel comfortable accepting the paper in its current form. The work would benefit from another round of review after significant revisions based on reviewer comments.

**Justification For Why Not Lower Score:**

N/A.

**Metareview: Summary, Strengths And Weaknesses:**

This paper presents some interesting empirical studies on math word problem solving. The authors compared two supervision methods – one with only final answers as the supervision signals and the other assuming the intermediate answers are available as supervision signals.

The reviewers largely agree this work focuses on a meaningful direction, is well-written, and comes with significant empirical findings that can be worth sharing with the community.

However, one concern with the current draft is that the work needs to include further insights by answering key questions underlying the observations made. For example, the authors argue that both approaches can lead to almost the same level of empirical success. Although I believe this is an important finding and have no doubt about its empirical validity, I expect to see why this is the case. Or, if there are challenges with explaining this, it would be good to have some in-depth discussions or careful analyses of such a finding when sharing it with the community. We could regard the intermediate steps as latent variables in the setup where there are only final answers as supervision signals. The quality of such latent variables will play a crucial role in the final performance. Could one possible reason be that the intermediate steps that the language model can produce (potentially with prompts) tend to be highly similar to those written by humans (in the process-based setup)? Including further insights would be beneficial in strengthening the work. There are other concerns about the stability of the results, given there are various hyperparameters and the authors only provided results over one dataset in their initial submission. Overall, the current draft is not yet ready and would benefit from further revisions based on the reviewers' comments.